# An improved ant colony optimization algorithm based on context for tourism route planning

**Shengbin Liang**[1], **Tongtong Jiao**[1], **Wencai Du**[2], **Shenming Qu**[1]*

**1** School of Software, Henan University, Kaifeng, Henan, China, **2** Institute for Data Engineering and Science, University of Saint Joseph, Macau SAR, China

* qsm@vip.henu.edu.cn

**Data Availability Statement:** All relevant data are within the manuscript.

**Funding:** Author: Liang Shengbin Grant number: No.20A520008 Funder: Key Scientific Research Projects of Colleges and Universities in Henan

## Abstract

To solve the problem of one-sided pursuit of the shortest distance but ignoring the tourist experience in the process of tourism route planning, an improved ant colony optimization algorithm is proposed for tourism route planning. Contextual information of scenic spots significantly effect people's choice of tourism destination, so the pheromone update strategy is combined with the contextual information such as weather and comfort degree of the scenic spot in the process of searching the global optimal route, so that the pheromone update tends to the path suitable for tourists. At the same time, in order to avoid falling into local optimization, the sub-path support degree is introduced. The experimental results show that the optimized tourism route has greatly improved the tourist experience, the route distance is shortened by 20.5% and the convergence speed is increased by 21.2% compared with the basic algorithm, which proves that the improved algorithm is notably effective.

## Introduction

With the development of mobile Internet technology, the user travel information is becoming more and more diversified. Travel notes, strategies, short videos and so on have become the main basis for the user tourism route planning.

While tourism route planning is a kind of loop problem that tourists start from the starting point, pass through the selected scenic spots only once, and finally return to the starting point. Similar to (Travelling Salesman Problem)TSP, tourism route planning is a NP-Hard problem, but it is more complex and affected by many constraints. When tourists arrive in a city to visit the scenic spots in its territory, they are often affected by time, transportation conditions, economic cost and other factors. In practice, people often spend a lot of time on the selection of scenic spots and route planning.

ACO algorithm is proposed by Dorigo et al. [1] according to the intelligent behavior of ant colony in the process of foraging. The algorithm has some advantages such as heuristic, positive feedback and distributed. ACO algorithm has been widely used to solve (Traveling Salesman Problem)TSP. Stutzle et al. [2] proposed Max-Min ant system (MMAS) to solve the problem but the improved algrithm is easy to fall into local optimum; Yang et al. [3] proposed

Province URL: http://jyt.henan.gov.cn/2019/07-31/1604951.html Did the sponsors or funders play any role in the study design, data collection and analysis, decision to publish, or preparation of the manuscript? No.

**Competing interests:** The authors declare that no competing interests exist.

an improved ACO algorithm based on game theory, which introduced entropy weighted learning strategy to optimize the accuracy of the optimal solution of (Traveling Salesman Problem)TSP problem on the basis of ACO and MMAS. Deng et al. [4] divided the optimization problem into several sub problems in order to improve the convergence rate of ACO algorithm and the pheromone update strategy was used to improve the optimization ability, then coevolution mechanism was used to exchange information among different sub populations, so as to avoid the ant colony falling into local optimum. Qian et al. [5] took the traffic cost as the calculation object, optimized the automatic adjustment mechanism of ant colony pheromone, and improved the performance of ACO algorithm. In addition, some researchers combined ACO algorithm with other algorithms. For example, Liang et al. [6] combined genetic algorithm with ACO algorithm to further improve the ability to solve the defects of local optimization, obtain the best path and save cost. Che et al. [7] combined particle swarm optimization with ACO algorithm to find the optimal path and improved the quality of path planning by improving pheromone update rules and heuristic function based on particle swarm optimization.

Scholars have put forward a large number of improvement methods and strategies to solve various problems in different boundaries, different disciplines and different fields. Aiming at the problems of local optimization and slow convergence of ACO algorithm, Wang et al. [8] embedded genetic algorithm and cloud model into ACO algorithm to obtain the optimal solution. Han et al. [9] proposed an an algorithm called SOS-ACO that combines symbiotic organisms search (SOS) and ACO algorighm to calculate the optimal or near-optimal assembly sequence, which find the optimal or near optimal assembly sequence in fewer iterations. ACO algorithm is also widely used in feature selection. Zhou et al. [10] proposed a two-stage hybrid ACO for high-dimensional feature se-lection (TSHFS-ACO), which uses the interval strategy to determine the size of OFS for the following OFS search and helps to reduce the complexity of the algorithm and alleviate the algorithm from getting into a local optimum. In solving constraint satisfaction problem (CSP), in order to overcome the shortcomings of low solution quality and slow convergence speed based on ACO algorithm, Guan et al. [11] proposed an improved ant colony optimization with an automatic updating mechanism (AU-ACO). In order to take advantage of both epsilon greedy and Levy flight, Liu et al. [12] proposed a greedy-Levy ACO incorporating these two approaches to solve complicated combinatorial optimization problems, which outperforms max-min ACO and other latest solvers. It is useful to implement collective intelligence(CI) evolution using ant colony optimization (ACO), Gan et al. [13] analyzed the performance of ACO evolution algorithm and verified the feasibility of applying the collective intelligence (CI) evolution theory to a specific application.

Besides ACO algorithm, traditional route planning methods include tabu search algorithm [14], genetic algorithm [15, 16], particle swarm optimization algorithm [17], simulated annealing algorithm [18] etc. ACO algorithm is a parallel algorithm, the search process of each ant is independent, and ants communicate through pheromone. Therefore, ACO algorithm can be regarded as a distributed multi-agent system, which starts to search for independent solutions at multiple points in the problem space at the same time, which not only increases the reliability of the algorithm, but also makes the algorithm have strong global search ability. In this paper, a solution method of tourism route planning problem based on context feedback and ACO algorithm is proposed.

In the above studies, the shortest distance is used as the goal of route planning. However, in the tourism route planning, in addition to the main factors that affect the route planning, the waiting time (comfort degree of the scenic spot) and the weather also have an important impact on the route planning. In ACO algorithm, ants communicate with each other through pheromones, and use learning mechanism to adjust the selection probability of the optimal

path. However, the ACO algorithm also has some defects, such as the lack of pheromone in the initial stage of path construction, the evolution speed is very slow; with the ACO algorithm using positive feedback principle to strengthen the optimal solution, after a certain number of iterations, pheromone is mainly concentrated on a few routes, resulting in premature convergence. At the same time, from the perspective of adjusting the load of scenic spots, the routes to individual scenic spots also lead to a sharp increase in the number of tourists, resulting in a sharp decline in tourist experience and a longer waiting time.

We propose a tourism route planning method based on contextual information feedback mechanism and ant colony optimization algorithm in this paper. The major works involved in this paper are as follows:

1. Take distance, scenic spots comfort ratio, as comprehensive evaluation indicators to improve tourists experience while ensuring economic costs.

2. The tourism route should pass through all the scenic spots selected by tourists, only once, and finally return to the starting point.

3. Introduce the comfort degree of the scenic spot, weather and other contextual information into the improved algorithm, and dynamically adjusts the tourism route planning through self-learning.

4. From the perspective of tourists, the tourism route planning should be people-oriented, reduce the economic cost of tourists and time, enhance the travel experience of tourist.

This paper proceeds in the following order. In section II, introduce ACO algorithm and its lastest researches. In section III, the proposed ACOCA algorithm is described in detail, including contextual information modelling, and ACOCA algorithm description. In section IV, a series of performance tests are simulated and the results of the experiments are analyzed. In section V, the conclusions and our further research directions are given.

## ACO algorithm

ACO algorithm is a kind of swarm intelligence algorithm. By simulating the process of ant colony foraging, the shortest path in different environments is established by using the internal information transmission mechanism of ant colony. In the process of activity, ants will leave pheromone, and the subsequent ants can choose the path according to the pheromone left by the previous ants. The higher the concentration of pheromone remaining on the path, the higher the probability that the ant will choose this path. At the same time, the concentration of pheromone will volatilize with time. Therefore, through the ant colony behavior, ants continuously learn and optimize through the information feedback mechanism to determine the shortest foraging path. According to the characteristics of ACO algorithm, it has been widely used in path planning [19], network routing [20], logistics distribution [21], trip route planning [22] and traveling salesman problem [6, 23].

For the studies of the ACO algorithm, some researchers proposed a lot of improved methods and applications in recent years. Yang et al. [24] proposed an improved ant colony optimization algorithm to control an output scattered light field after an incident beam of light passes through a turbid medium. Yu et al. [25] proposed an ant colony algorithm based on magnetic neighborhood and filtering recommendation (MRACS) to deal with the problems that the ACO algorithm has slow convergence speed and easy falling into the local optimum when solving (Traveling Salesman Problem)TSP. Wu [26] proposed a hybrid ant colony optimization (HACO) algorithm for solving the problem of vehicle routing with time windows. In view of the shortcomings of traditional ACO algorithm in path planning of indoor mobile robot,

such as a long time path planning, non-optimal path for the slow convergence speed, and local optimal solution characteristic of ACO, Miao et al. [27] proposed an improvement adaptive ant colony optimization (IAACO) algorithm to address these problems. Jia et al. [28] proposed an ant colony-based algorithm for integrated scheduling on batch machines with non-identical capacities. Cerda et al. [29] proposed an algorithm based on ACO algorithm metaheuristics to dynamically optimize the decision thresholds provided by the Pairs Trading investment strategy. The control method of space vector pulse width modulation (SVPWM) overmodulation region II has the disadvantages of a complicated process and large harmonic content. To solve these problems, Zhang [30] proposed an SVPWM overmodulation region II control method based on the chaos ant colony algorithm. To address the lack of convergence speed and diversity of ACO algorithm, Yu et al. [31]proposed a dynamic reproductive ant colony algorithm based on piecewise clustering (RCACS) to optimize the problems. Xiao et al. [32] proposed an improved ACO algorithm with a negative feedback mechanism to adress the ACA encounters difficulty escaping from the local optimal solution. Based on ant colony optimization algorithm, Wang et al. [33]constructs a sports competition venue simulation method based on ant colony optimization algorithm and artificial intelligence to improve the reduction degree of sports competition venue simulation. Kanso et al. [34] proposed an Hybridized Ant Colony Algorithm (HACA) and hybridized with a local search algorithm that involves the 2-opt, Swap, Relocate and Cross-exchange moves to solve OCARP problem. Shi [35] proposed a routing protocol and simulated based on an ant colony optimization algorithm's performance for reducing the energy consumption of sensors in IoT networks for increasing network lifespan.

As a meta-heuristic algorithm, ACO algorithm is a candidate method in supply chain management (SCM). According to Axsater [36, 37] and Mostafa et al. [38], the cost functions of two or three-echelon inventory systems were proposed. Based on (R, Q) ordering policy, Mostafa et al. [39] showed the information sharing's influence on inventory costs. With the development of meta-heuristic algorithms, several different meta heuristic algorithms have been used to solve the two-tier supply chain problem [40], improve the efficiency of the supply chain [41], solve the vehicle routing problem [42], and optimize the response time in medical treatment [43].

The above methods confirm that ACO algorithm has the advantages of strong adaptability and dynamic optimization through feedback mechanism. However, it also has the defects of long convergence time and the inter-colony communication strategy is not delicate enough to ensure the adaptability of ACO, so it is easy to fall into local optimization.

The basic idea of ACO algorithm is illustrated by an example of tourist route planning. There are m scenic spots, $d_{ij}(i, j = 1, 2. . ., m)$is the distance between scenic spot i and j, $\tau_{ij}(t)$ represents the amount of information between scenic spot i and j, at time t, that is, the pheromone of simulated ants. Let there be n ants in total, $p_{ij}^k(t)$ is the probability of ant k from scenic spot i to j at time t, then Eq (1):

$$p_{ij}^k(t) = \begin{cases} \dfrac{\tau_{ij}^{\alpha}(t)\eta_{ij}^{\beta}(t)}{\sum_{\gamma \in allowed_k} \tau_{ir}^{\alpha}(t)\eta_{ir}^{\beta}(t)}, & j \in allowed_k \\ \\ 0, & otherwise \end{cases} \tag{1}$$

Among them, $allowed_k$ is the collection of scenic spots allowed to visit in the next step of ant k. $\alpha$ represents the importance of pheromones on the path to the ants' selection path, $\eta_{ij}$ is the expected degree of transferred from scenic spot i to j, $\beta$ is the degree of importance to $\eta_{ij}$. When $\beta = 0$, it is an heuristic algorithm representing positive feedback, and it is a greedy algorithm in essence when $\alpha = 0$. Generally, $\eta_{ij}$ is closely related to the distance between the two

scenic spots, generally $\eta_{ij}$ is calculated by Eq (2).

$$\eta_{ij} = \frac{1}{d_{ij}} \tag{2}$$

Then, after each ant visits all the scenic spots, the pheromone is updated. The update rules are as follows(Eqs (3), (4) and (5)):

$$\tau_{ij}(t+1) = \rho\tau_{ij} + \Delta\tau_{ij}(t, t+1) \tag{3}$$

$$\Delta\tau_{ij}(t, t+1) = \sum_{k=1}^{n} \Delta\tau_{ij}^{k}(t, t+1) \tag{4}$$

$$\Delta\tau_{ij}^{k} = \begin{cases} \frac{Q}{L_k}, & when\ kth\ ant\ visit\ i\ from\ j \\ 0, & otherwise \end{cases} \tag{5}$$

$\Delta\tau_{ij}^{k}(t, t+1)$ represents the amount of information that ant k stays on the road from scenic spot i to j at the time of $(t, t+1)$, $\Delta\tau_{ij}(t, t+1)$ represents the increment of pheromone in scenic spot i to j, and $\rho$ is the change system of pheromone in the section with the value between (0,1). Q is a constant, $L_k$ is the length of the path taken by ant k.

## Proposed method

Traditional tourism route planning methods introduce some heuristic algorithms to solve the optimal path, and achieves good results. Huanwg introduced the chance algorithm into ACO, which has good effect in dynamic tourism route planning [22]. Mei took time as the key constraint, and gave a tourism route planning prototype combined with ant colony algorithm [44]. However, the above approaches suffer from two limitation when handling tourism route planning. One limitation is that pheromone updating mechanism leads people to gather in a scenic spot, which brings great challenges to tourism management and service. Another limitation is that the accuracy of these approaches in large-scale tourism route planning problem is not satisfactory. We propose a tourism route planning algorithm based on the integration of environment feedback mechanism. The algorithm is based on ACO algorithm integrated context aware(ACOCA) and introduces the contextual information feedback mechanism. It introduces some contextual information such as comfort degree of the scenic spot, weather and other contextual data into the algorithm, in order to improve the pheromone update method to obtain the optimal tourism route on the basis of ensuring the user experience of tourists.

The main strategy of ACOCA algorithm: for the contextual information of the target scenic spot, if it is beneficial to the tour, increase the pheromone concentration on the path to the scenic spot, otherwise reduce it; At the same time, in order to avoid ant colony falling into local optimization, the sub-path support degree of the sub-path is less than the threshold, the algorithm will enter the next iteration.

### Contextual information and representation

The contextual information of scenic spots, especially the weather and the number of tourists significantly affect the willingness of tourists to travel. For weather information, different scenic spots have different requirements for weather. For example, outdoor scenic spots are more sensitive, while indoor tourism projects have lower requirements for weather. The factor

*Weather$_i$* of weather condition on scenic spot i, its vaule range is in the range of [0, 1], where 1 is strongly recommended and 0 is not recommended.

The number of tourists varies with the capacity of different scenic spots, so the comfort degree of scenic spot is used to express the crowding intensity of scenic spot i is obtained by Eq (6).

$$Comf_i = N_i / N_{max} \tag{6}$$

Where $N_i$ is the current tourists number of scenic spot i, $N_{max}$ is the maximum tourist number scenic spot i can receive. *Comf$_i$* is devided into four ranges, [0,0.25] means comfortable, (0.25,0.50] stands for general comfortable, (0.50,0.75] represents average and (0.75,1] is crowded. According to the management experience of the scenic spot, when the comfort degree of the scenic spot is greater than 0.75, means that tourists are no longer recommended to rush into the scenic spot.

## Contextual distance

$d_{ij}(i, j = 1, 2. . ., m)$ is the physical distance between the two scenic spots. When calculating the physical distance between two scenic spots, the earth is calculated as an approximate sphere. By obtaining the longitude and latitude coordinates of scenic spots, the calculation method of distance is shown in Eq (7):

$$d_{ij} = R \times \arccos(\sin(Lat_i * \frac{\pi}{180}) \times \sin(Lat_j * \frac{\pi}{180}) + \cos(Lat_i * \frac{\pi}{180})$$
$$*\cos(Lat_j * \frac{\pi}{180}) * \cos(|Lon_i - Lon_j| * \frac{\pi}{180})) \times \frac{\pi}{180} \tag{7}$$

where R is the radius of the earth, and the average radius is 6371.004km. The latitude and longitude coordinates of scenic spot i are expressed by $(Lon_i, Lat_i)$.

It should be pointed out that considering the contextual information of the scenic spot, if the number of tourists in scenic spot i is too large at a certain time, tourists are not recommended to visit the scenic spot again; if the number of tourists in scenic spot j is appropriate, tourists are recommended to visit the scenic spot j, then the contextual distance between scenic spots $Distance_{ij} \neq Distance_{ji}$(Eq (8)).

$$Distance_{ij} = \begin{cases} d_{ij} * \sqrt{Out_i}, & Comf_j \leq 0.75 \\ d_{ij} * \sqrt{\frac{Out_i}{Comf_j}}, & otherwise \end{cases} \tag{8}$$

The outdegree $Out_i$ of departure scenic spot i is introduced to calculate the contextual distance. It means that when the destination scenic spot j is not suitable for sightseeing, the larger of outdegree $Out_i$, the higher probability of the scenic spot i to other scenic spots other than the target scenic spot j.

## Context feedback mechanism

Under the background of smart tourism construction, tourists can obtain the contextual data of scenic spot in real time and make relevant preparations in advance. In ACO algorithm, the only criterion for each ant to determine the path is pheromone concentration, but because of the single criterion, it is easy to lead to uneven number of people between scenic spots. Therefore, we introduce the combination of contextual data and pheromone as the criteria of ant colony path selection. We propose an ant colony optimization integrated with context aware algorithm, which is based on ACO algorithm and optimizes the pheromone updating rules

and transition probability of traditional ACO algorithm, so as to make the contextual information of scenic spots participate in pheromone updating. The main principles of context information collaborative improvement pheromone update are as follows:

1. When the weather and comfort degree of the scenic spot are not suitable for tourists, it is necessary to artificially stimulate all route pheromones to the scenic spot according to the contextual information of the scenic spot, and reduce the pheromone concentration of these routes when updating the pheromones.

2. The pheromone update of basic ACO algorithm is positive feedback. Positive feedback is conducive to the convergence of the algorithm, but it also reduces the diversity of the population, resulting in the ant colony concentrated on a few routes, resulting in traffic pressure and overload of scenic spots. In order to avoid the premature convergence of ant colony to form the local optimal solution, the route selection probability is modified to make up for the premature phenomenon caused by the disorder and irregular search of ACO algorithm.

Therefore, based on the above principle 1 and improved Eq (2), context distance is introduced to replace physical distance(Eq (9)):

$$\eta_{ij} = Weather_j / Distance_{ij} \qquad (9)$$

In the Eq (9), the heuristic information of $route(i, j)$ is determined by the weather, comfort degree of scenic spot and physical distance $d_{ij}$. Positive feedback when the weather of the target scenic spot j is not suitable for sightseeing; positive feedback when the comfort degree is suitable for sightseeing, otherwise negative feedback.

According to the above principle 2, because the pheromone updating strategy in basic ACO algorithm only occurs on the optimal route, and the sub route of the optimal route may be longer, the ants choose the shortest route in the optimal route, which is the main reason for premature. The concept of subpath support degree(SPSD) is introduced, and the calculation formula is shown in Eq (10). Among the optimal solutions found by each generation of ants, the routes whose support degree of $route(i, j)$ to the overall optimal path is greater than the route contribution threshold are found, and the pheromones of $route(i, j)$ are enhanced and updated, and the update formula is shown in Eqs (11) and (12).

$$SPSD_{ij} = \frac{Distance_{ij}}{L_i} \qquad (10)$$

$$\Delta\tau ij' = \begin{cases} \frac{Q}{Distance_{ij}}, & when \ SPSD_{ij} > x \\ 0, & otherwise \end{cases} \qquad (11)$$

$$\tau_{ij}(t + 1) = \Delta\tau ij' + \tau_{ij}(t + 1) \qquad (12)$$

Where $\tau_{ij}(t + 1)$ is still calculated by Eq (3), x is the route support threshold, when $SPSD_{ij}$ is greater than the threshold, the pheromone of the sub $route(i, j)$ is updated, and Q is the pheromone increase coefficient, that is, the total amount of pheromones released by ants in an iteration.

## ACOCA algorithm description

The basic goal of ACOCA algorithm is that the planned route is an optimal solution: firstly, the proportion of comfort passing through the scenic spots is low, and secondly, the path

distance is the shortest. According to the contextual information of scenic spots and path distance, the cost function is obtained, as shown in Eq (13), where makes the path cost V the lowest, $s_{ij}$ is the comfort degree of the scenic spot i to j, and $d_{ij}$ is the distance from scenic spot i to j. The constraints are expressed by Eqs (14)–(17), Where $s_{k_{ij}}$ and $y_{k_i}$ are calculated by Eqs (18) and (19).

$$minV = \sum_{k=1}^{m}\sum_{j=1}^{n}\sum_{i=1}^{n} s_{ij}d_{ij} \tag{13}$$

$$s.t\sum_{k=1}^{m} y_{k_i} = 1, i = 2, 3, ..., n \tag{14}$$

$$\sum_{i=2}^{n} s_{k_{ij}} = y_{k_i}, j = 2, 3, ..., n; k = 1, 2, ..., m \tag{15}$$

$$\sum_{j=2}^{n} s_{k_{ij}} = y_{k_i}, i = 2, 3, ..., n; k = 1, 2, ..., m \tag{16}$$

$$\sum_{i=2}^{n} y_{k_i} c_i \leq C_{max}, k = 1, 2, ..., m \tag{17}$$

$$s_{k_{ij}} = \begin{cases} 1, & user\ k\ from\ scenic\ spot\ i\ to\ j \\ 0, & otherwise \end{cases}, i, j = 2, 3, ..., n; k = 1, 2, ..., m \tag{18}$$

$$y_{k_i} = \begin{cases} 1, & scenic\ spot\ i\ meets\ user\ k \\ 0, & otherwise \end{cases}, i = 2, 3, ..., n; k = 1, 2, ..., m \tag{19}$$

The pheromone is updated through the contextual information of scenic spots, the basic workflow as following(Fig 1).

## Experimental evaluation

### Datasets and parameter settings

In order to test the performance of ACOCA algorithm in route planning in different regions, we adopt three groups of scenic spots with different geographical distribution, as shown in Fig 2 below.

All of the algorithms implement in Python, we run the proposed algorithm and the base algorithm 10 times in each dataset, and calculate the average of the experimental results for comparison. The parameters settings in our experiments of ACOCA as Table 1, and the experimental environment shown as Table 2.

### Evaluation protocol

We mainly evaluate from the following three aspects: total route distance, convergence time and user comfort ratio.

Total route distance: From the perspective of economic cost, the total distance of the route is evaluated. The tourism practice route planning algorithm plans the total distance of the

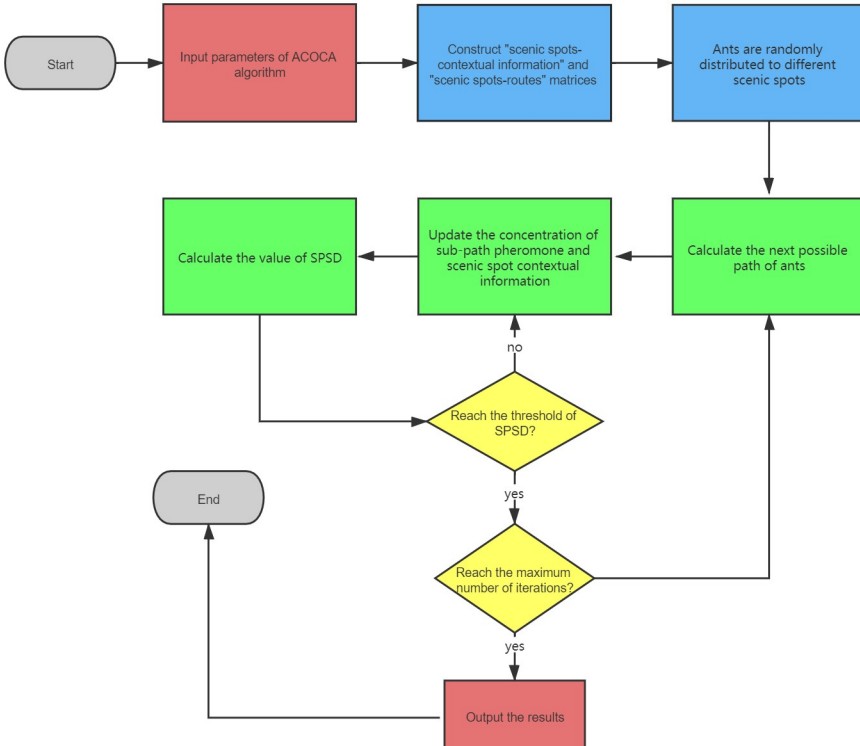

**Fig 1. The workflow of ACOCA algorithm.** Input: set the parameters of the algorithm, such as the number of ants, the maximum number of iterations, the initial comfort degree of each scenic spot, and the local optimal threshold x. Output: the optimal solution and its route set. Step 1: randomly put the ants in different scenic spots as the starting point; Step 2: each ant calculates the solution space and the comfort degree of each scenic spot; Step 3: calculates and updates the pheromone concentration on the connection path between scenic spots according to Eqs (5) and (9); Step 4: updates the comfort degree of the scenic spot every iteration; Step 5: judge whether the algorithm falls into local optimum. Judgment method: compare the global optimal solution obtained in this iteration with the global optimal solution obtained in the previous iteration. If the continuous x-times optimal solution is not improved, it is considered to fall into local optimum and implements the local optimal strategy; otherwise, it enters step 6; Step 6: determines whether the maximum number of iterations is reached. If so, the algorithm ends, otherwise, step 1 is executed again.

route from the starting point to the starting point after visiting several scenic spots in a certain order.

Convergence time: From the point of view of the performance of the algorithm, the convergence time is evaluated. Because ACOCA and the base algorithms are heuristic algorithms, the algorithm takes a finite number of iterations to get a stable solution, that is, the convergence time represents the performance of the algorithm.

User comfort ratio: From the perspective of tourist user experience, the scenic comfort degree is evaluated. It mainly measures the tourists' comfort in the planned route, such as the scenic spot is crowded or not, the weather of the scenic spot is suitable for visiting or not. The evaluation is based on the number of crowded scenic spots in the whole journey.

## Experimental results

The running results of ACOCA algorithm and base algorithms in three groups of scenic spots with different geographical distribution are given in Table 3. The results are the average of each algorithm running 10 times in each group of data. The comfort ratio is the percentage of scenic spots with comfort degree higher than 0.75 in all scenic spots of the planned route(the higher the comfort ratio, the lower the overall user satisfaction).

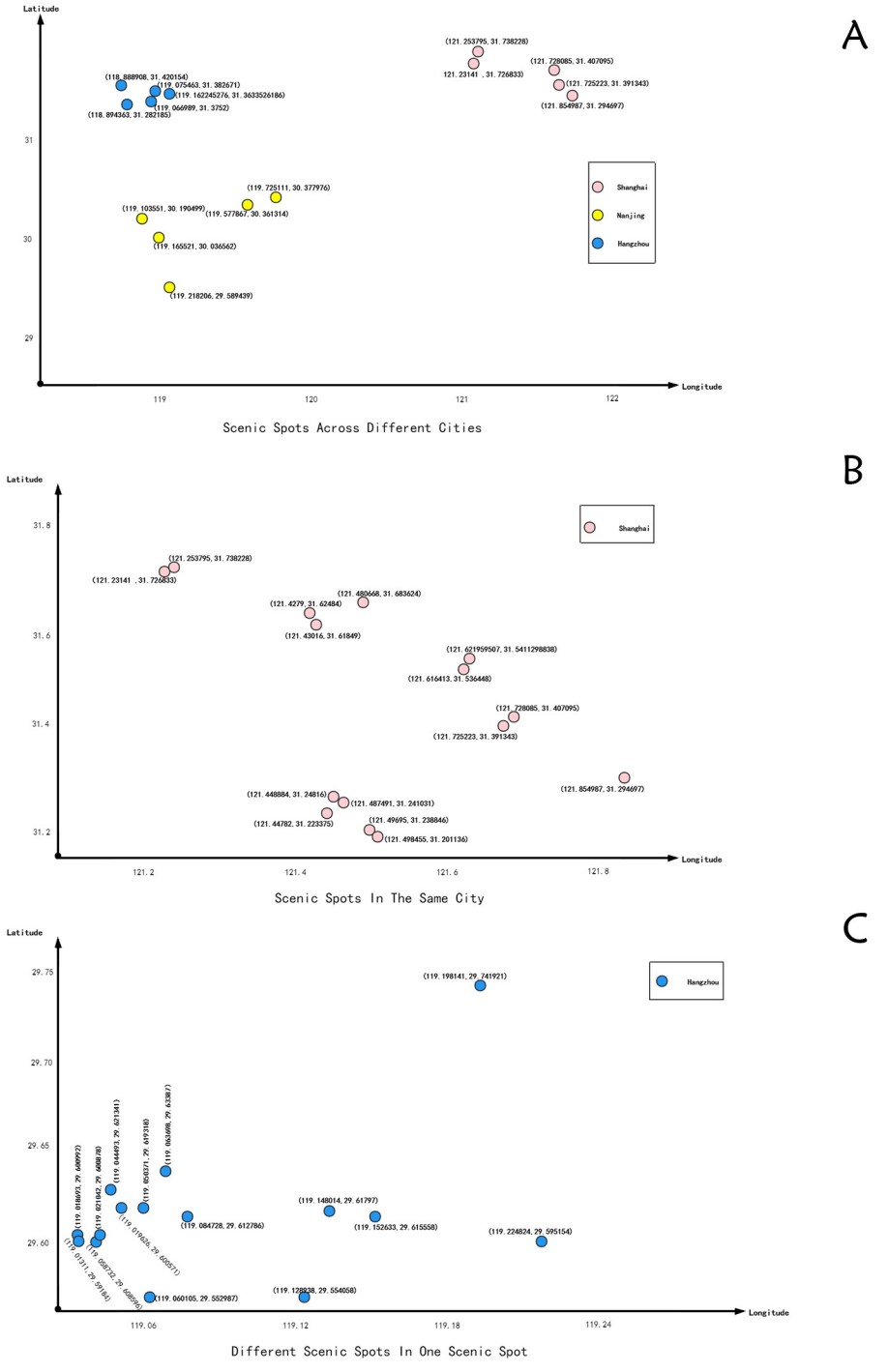

**Fig 2. Scenic spots.** A: scenic spots across different cities, including 15 scenic spots in Shanghai, Hangzhou and Nanjing of China. B: scenic spots in the same city, including 15 scenic spots in Shanghai. C: different scenic spots in one scenic spot, we take 15 scenic spots of Qiandao Lake.

**Table 1. Parameters settings.**

| Ant Numbers | Maximum Iterations | Initial Pheromone | $\alpha$ | $\beta$ | $\rho$ |
|---|---|---|---|---|---|
| 20 | 100 | 0.5 | 1.0 | 4.5 | 0.1 |

**Table 2. Experimental environment.**

| hardware and software | values |
|---|---|
| CPU | Intel i5 @ 2.4GHz |
| RAM | 8GB |
| OS | Windows 10 |
| IDE | PyCharm in Python 3.7 |

It can be seen from Figs 3 and 4 that the comfort ratio is obviously better than the ACO algorithm by introducing the comfort degree and weather information, which improves the user's tourism experience. Through real-time contextual information, it can help users to plan tourism routes better. When the distance between scenic spots becomes larger, the optimal path of ACOCA algorithm has some advantages over ACO algorithm. By introducing the support degree of path, the local optimal problem is effectively solved, and the solution of the problem has a good performance. Therefore, on the premise that the efficiency of the algorithm is almost the same, ACOCA algorithm can better improve the tourists' travel experience. At the same time, the distance of the travel route planned by ACOCA algorithm is also better than that of ACO algorithm.

**Table 3. The results of experiments.**

| Different Dataset | Average Distance | Average Convergence Time | Average Comfort Ratio |
|---|---|---|---|
| ACOCA(same city) | 203.4739 | 2.2676 | 26% |
| ACOCA(different cities) | 885.7403 | 2.3338 | 27% |
| ACOCA(same scenic spot) | 70.6389 | 2.3286 | 28% |
| ACO(same city) | 205.7595 | 3.0594 | 73% |
| ACO(different cities) | 892.3387 | 2.5736 | 88% |
| ACO(same scenic spot) | 71.7929 | 3.0335 | 78% |

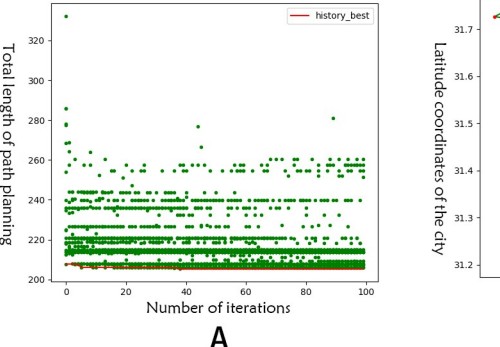 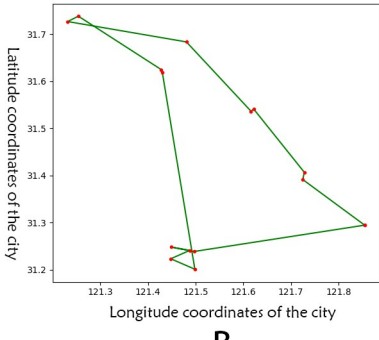

A B

**Fig 3. Performance of ACOCA and ACO.** From the comparison results of the two algorithms in different datasets, the distance of ACOCA algorithm is better than that of ACO algorithm.

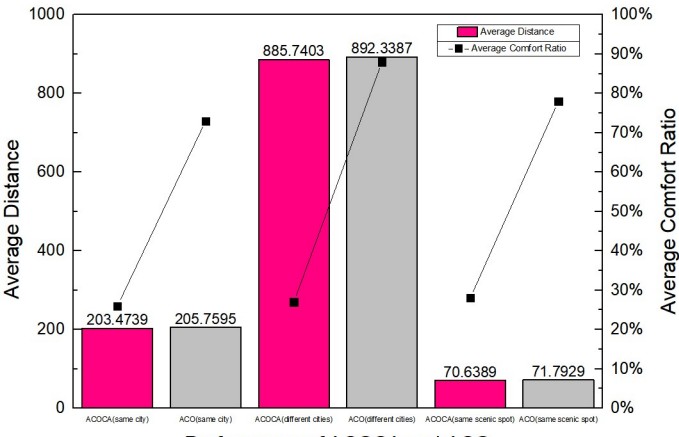

**Fig 4. Comparison of two algorithms in convergence time.** From the comparison results of the two algorithms in different datasets, the convergence time of ACOCA algorithm is better than that of ACO algorithm.

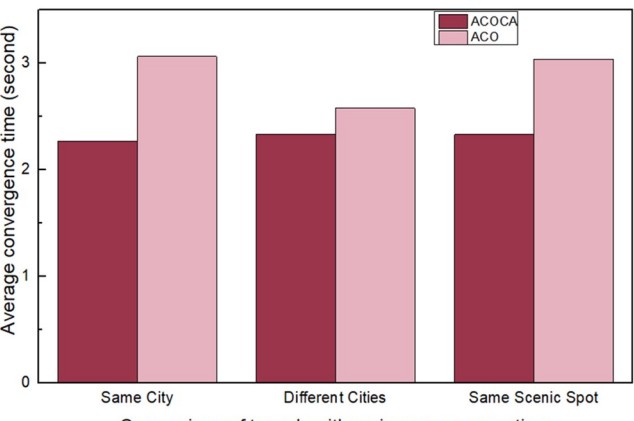

**Fig 5. ACOCA algorithm in the same city planning results and convergence.** A: The convergence of ACOCA algorithm. B: The result of ACOCA algorithm.

Taking the route planning results of ACOCA algorithm in the same city as an example, Fig 5 shows the result of ACOCA algorithm and its convergence. The results show that the algorithm is stable after the 40th iteration, and the convergence effect is good, and the planning route is in line with the actual context.

At the same time, in order to verify the universality of ACOCA algorithm, experiments are carried out on the open dataset TSPLIB st70, and the experimental results are shown in Table 4.

**Table 4. The results of experiments.**

| Algorithm | Distance(st70) | Time(st70) | Distance(different cities) | Time(different cities) |
|---|---|---|---|---|
| ACO | 767.7446 | 94.1582 | 892.3387 | 2.5736 |
| GA | 979.5677 | 20.1540 | 897.3524 | 1.2420 |
| Hybrid-ACO | 884.2793 | 95.6306 | 976.4651 | 4.2887 |
| ACOCA | 703.0122 | 75.3935 | 885.7403 | 2.3338 |

It can be seen from Table 4 that the ACOCA algorithm proposed in this paper has great advantages in solving the distance of the optimal path on two datasets, especially in the large-scale dataset st70, the ACOCA algorithm is 9.21% shorter than the ant colony algorithm, 25.78% shorter than the Hybrid-ACO algorithm, and has obvious advantages in planning the shortest path distance. But genetic algorithm (GA) is a global solution, its convergence time is incomparable to ACO algorithm. In addition, the four algorithms in the same city of 15 scenic spots path planning, ACOCA algorithm in the planning path distance is also the best, and the convergence time is not different from GA algorithm, indicating that ACOCA algorithm performs better in small-scale datasets.

## Conclusion and future work

In order to optimize the tourism route planning problem, in this paper, an improved ant colony optimization(ACOCA) algorithm based on context-aware mechanism, pheromone updating strategy is proposed to balance route distance and user comfort degree. In the proposed ACOCA algorithm, we introduce the weather and comfort degree of scenic spots to decide the pheromone updating strategy, and avoid to fall into the local optimum value, the concept of sub-path support degree was set. In order to verify the optimization performance of the ACOCA algorithm, We select three groups of scenic spots with different regional ranges, each group includes 15 scenic spots, and make route planning for each group of scenic spots; At the same time, we also carried out simulation experiments on TSPLIB dataset. The experimental results show that the performance of ACOCA algorithm is compared with basic ACO algorithm, genetic algorithm and Hybrid-ACO algorithm on the two datasets, and its operation results are optimized in route distance, convergence time and user comfort ratio. In particular, compared with Hybrid-ACO algorithm, the travel route distance is reduced by 20.5% and the convergence time is reduced by 21.2%, At the same time, user travel comfort has also been greatly improved. ACOCA algorithm has better optimization ability and user comfort than the baseline algorithms.

However, the efficiency of ACOCA algorithm needs to be further improved when the scale of scenic spots increases greatly. At the same time, we plan to include another contextual factors such as tourism cost into the algorthm. In the future work, the ACOCA algorithm will be studied deeply.

## Acknowledgments

The author would like to thank the editor and the anonymous reviewers for their valuable comments.

## Author Contributions

**Conceptualization:** Shengbin Liang.

**Data curation:** Tongtong Jiao.

**Formal analysis:** Wencai Du, Shenming Qu.

**Funding acquisition:** Shengbin Liang, Wencai Du.

**Investigation:** Wencai Du.

**Methodology:** Shengbin Liang.

**Project administration:** Tongtong Jiao.

**Resources:** Tongtong Jiao.

**Software:** Shengbin Liang, Tongtong Jiao, Shenming Qu.

**Supervision:** Shengbin Liang, Tongtong Jiao, Shenming Qu.

**Validation:** Tongtong Jiao.

**Visualization:** Tongtong Jiao.

**Writing – original draft:** Shengbin Liang.

**Writing – review & editing:** Tongtong Jiao.

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
