## [Decision Letter · Decision Letter 0]

5 Aug 2021

PONE-D-21-21787

A tourism route planning algorithm based on context feedback mechanism

PLOS ONE

Dear Dr. Qu,

Thank you for submitting your manuscript to PLOS ONE. After careful consideration, we feel that it has merit but does not fully meet PLOS ONE’s publication criteria as it currently stands. Therefore, we invite you to submit a revised version of the manuscript that addresses the points raised during the review process.

ACADEMIC EDITOR:

The quality of the paper must be improved, in specific the novelty of the manuscript and the related literature.

We look forward to receiving your revised manuscript.

Kind regards,

Diego Oliva

Academic Editor

PLOS ONE

Journal Requirements:

2. We note that Figure 2 in your submission contain [map/satellite] images which may be copyrighted. All PLOS content is published under the Creative Commons Attribution License (CC BY 4.0), which means that the manuscript, images, and Supporting Information files will be freely available online, and any third party is permitted to access, download, copy, distribute, and use these materials in any way, even commercially, with proper attribution. For these reasons, we cannot publish previously copyrighted maps or satellite images created using proprietary data, such as Google software (Google Maps, Street View, and Earth). For more information, see our copyright guidelines: http://journals.plos.org/plosone/s/licenses-and-copyright.

1. You may seek permission from the original copyright holder of Figure 2 to publish the content specifically under the CC BY 4.0 license.  

Reviewers' comments:

Reviewer's Responses to Questions

**Comments to the Author**

1. Is the manuscript technically sound, and do the data support the conclusions?

Reviewer #1: Yes

Reviewer #2: Partly

2. Has the statistical analysis been performed appropriately and rigorously? 

Reviewer #1: Yes

Reviewer #2: Yes

3. Have the authors made all data underlying the findings in their manuscript fully available?

Reviewer #1: Yes

Reviewer #2: Yes

4. Is the manuscript presented in an intelligible fashion and written in standard English?

Reviewer #1: Yes

Reviewer #2: Yes

5. Review Comments to the Author

Reviewer #1: - First, the title of the paper is not suitable. I think you must use your novelties and the overall image of your paper.

- I encourage you to add more detail about your core contributions in the abstract.

- Literature review is very short and old! You have not covered the knowledge edge! Please clarify the contribution of the paper according to the research gap.

- Many recent papers in the area can be added to the literature review. I do not propose you a especial reference due to reviewing ethical issues.

- Please, clarify which one of the assumptions is new in this area in the problem definition.

- Check the English presentation of this paper to remove the typos mistakes.

- Findings, limitations, and recommendations of this paper can be discussed more in the conclusion section.

- Please draw and bring some better figures in terms of color and quality.

Reviewer #2: This paper deal with the new modification in ACO algorithm to solve a basic tourism route planning. The authors claimed that this new approach has enhanced the process of ACO to solve the problem. However, still some works must be done for better illustration of the whole manuscript. Please address the following comments and concerns:

6. PLOS authors have the option to publish the peer review history of their article (what does this mean?). If published, this will include your full peer review and any attached files.

Reviewer #1: No

Reviewer #2: No

---

## [Author Response · Author response to Decision Letter 0]

25 Aug 2021

Date: August 25, 2021

Dear editor and dear reviewers:

Thank you for giving us the opportunity to submit a revised draft of the manuscript “A tourism route planning algorithm based on context feedback mechanism” for publication in the PLOS ONE. We appreciate the time and effort that the reviewers and editors dedicated to providing feedback on our manuscript and grateful for the insightful comments on and valuable improvements to our paper.

We have incorporated most of the suggestions made by the reviewers. Those changes are highlighted within the manuscript. For more information, please check the manuscripts named 'Revised Manuscript with Track Changes' and 'Manuscript'.

If you have any question about this paper, please don’t hesitate to let me know.

 Sincerely yours,

---

## [Decision Letter · Decision Letter 1]

31 Aug 2021

An improved ant colony optimization algorithm based on context for tourism route planning

PONE-D-21-21787R1

Dear Dr. Qu,

We’re pleased to inform you that your manuscript has been judged scientifically suitable for publication and will be formally accepted for publication once it meets all outstanding technical requirements.

Kind regards,

Diego Oliva

Academic Editor

PLOS ONE

Additional Editor Comments (optional):

Reviewers' comments:

Reviewer's Responses to Questions

**Comments to the Author**

1. If the authors have adequately addressed your comments raised in a previous round of review and you feel that this manuscript is now acceptable for publication, you may indicate that here to bypass the “Comments to the Author” section, enter your conflict of interest statement in the “Confidential to Editor” section, and submit your "Accept" recommendation.

Reviewer #1: All comments have been addressed

Reviewer #2: All comments have been addressed

2. Is the manuscript technically sound, and do the data support the conclusions?

Reviewer #1: Yes

Reviewer #2: Yes

3. Has the statistical analysis been performed appropriately and rigorously? 

Reviewer #1: Yes

Reviewer #2: Yes

4. Have the authors made all data underlying the findings in their manuscript fully available?

Reviewer #1: Yes

Reviewer #2: Yes

5. Is the manuscript presented in an intelligible fashion and written in standard English?

Reviewer #1: Yes

Reviewer #2: Yes

6. Review Comments to the Author

Reviewer #1: The paper addressed all my concerns and can be accepted and published in this journal.

Reviewer #2: The authors were so exact on answering my comments and the answers were comprehensive. I suggest publication at this point.

7. PLOS authors have the option to publish the peer review history of their article (what does this mean?). If published, this will include your full peer review and any attached files.

Reviewer #1: No

Reviewer #2: No

---

## [Editor Report · Acceptance letter]

6 Sep 2021

PONE-D-21-21787R1 

An improved ant colony optimization algorithm based on context for tourism route planning 

Dear Dr. Qu:

I'm pleased to inform you that your manuscript has been deemed suitable for publication in PLOS ONE. Congratulations! Your manuscript is now with our production department. 

Kind regards, 

on behalf of

Dr. Diego Oliva 

Academic Editor

PLOS ONE